



# Can worms be used to produce amendments with reduced CO₂ emissions during co-composting with clay and biochar and after their addition to soil?

5    Justine Barthod [1], Cornélia Rumpel [1], Remigio Paradelo [2], Marie-France Dignac [1]

[1] Institute of Ecology and Environmental Sciences- Paris (iEES-Paris) UMR CNRS, INRA, UPMC. 78850 Thiverval-Grignon. France

[2] University of Vigo, Department of Plant Biology and Soil Sciences, Facultade de Ciencias, As Lagoas s/n, 32004 Ourense Spain

*Correspondence to:* Justine Barthod (justine.barthod@grignon.inra.fr)

**Abstract :**

In this study we evaluated $CO_2$ emissions during co-composting and co-vermicomposting of green wastes with clay and/or biochar. The stability of the final products as well as their effect on C mineralization in soil have been evaluated. The aim of the study was to test the following hypothesis: (1) interactions between clay and biochar and organic wastes would lead to reduced $CO_2$ emissions during the composting process, (2) these interactions would be enhanced in the presence of worms, and (3) more carbon would be sequestered in soil after the use of the resulting compost/vermicompost as amendments. We added two different doses of clay, biochar and their mixture to pre-composted green wastes and monitored C mineralisation during 21 days in presence or absence of worms (*Eisenia* species). The organic materials were then added to a loamy Cambisol and the $CO_2$ emissions were monitored during 30 days in a laboratory incubation.

Our results indicated that the addition of clay or clay/biochar mixture reduced carbon mineralization during co-composting without worms by up to 44%. However, in the presence of worms, $CO_2$ emissions increased for all treatments except for the low clay dose. The production conditions had more influence on C mineralization in soil for composts than for vermicomposts except for the low clay treatment, which showed a more reduced $CO_2$ emissions compared to a regular compost.

In summary, the addition of worms during co-composting with clay and biochar may be a promising technology for reducing $CO_2$ emissions and increasing soil carbon storage. We suggest that the production of a low $CO_2$ emission amendment requires optimisation of OM source, co-composting agents and worm species. The effect of the resulting material on soil fertility has to be evaluated.

**Keywords:** carbon mineralization; worm; composting; biochar; clay, soil.



## 1.Introduction

Industrial activity, deforestation and the utilization of fossil fuels are responsible for a steady increase of $CO_2$ in the atmosphere. In this context, massive soil organic matter (OM) loss is observed, leading to decline of many soil ecosystem services, such as fertility and carbon storage

(Smith et al., 2015). These global changes of the earth's climate and (agro-)ecosystems have major environmental, agronomic but also social and economic consequences, which could be attenuated by the rebuilding of soil OM stocks (IPCC, 2014). Increasing soil C may be possible with the use of composted organic wastes, which have been proposed as alternative fertilisers (Ngo *et al*, 2011, 2012), and which could counterbalance the concentration of greenhouse gases in the atmosphere

through soil C sequestration (Lashermes *et al*, 2009)

Two well-known aerobic processes based on microbial activity are able to transform organic wastes into valuable soil amendments: composting and vermicomposting. Composting has been traditionally used and leads to stabilized organic amendments with fertilization potential. During vermicomposting the presence of worms induces a continuous aeration resulting in a faster organic

matter transformation. However, vermicomposting and composting both emit greenhouse gases such as $CO_2$, $CH_4$ and $N_2O$ (Hobson *et al*, 2005; Chan *et al*, 2011; Thangarajan *et al*, 2013). In addition, the final products of these processes lead to greenhouse gas emissions after their application to soil (Cambardella *et al*, 2003, Bustamante *et al*, 2007). These emissions can originate from the mineralization of (vermi)compost OM itself or maybe due the mineralization of native soil

organic matter following increased microbial development and activity, a mechanism known as priming effect (Bustamante *et al*, 2010).

In order to optimize the recycling of waste C, there is a need to enhance OM stabilization during (vermi)composting. Stabilization mechanisms are poorly known for composting processes, while they have been widely studied in soils. Enhancing C stabilization in composts could thus benefit

from an analogy with the mechanisms known to occur in soils (v. Lützow *et al*, 2006): spatial inaccessibility, selective preservation due to chemical recalcitrance, and formation of organo-





mineral associations. Among these processes, the association of OM with minerals is the most

efficient for C stabilization on long time scales (Kleber *et al*, 2015). Therefore, a variety of minerals

has been used to reduce gas emissions ($CO_2$, $CH_4$, $NH_3$ and $N_2O$) during co-composting (Bolan *et*

*al*, 2012, Wang *et al*, 2014, Chowdhury *et al*, 2015), e.g. clay minerals during composting of

poultry manure (Bolan *et al*, 2012) or zeolite during (vermi)composting of wastes (Wang *et al*,

2014). However, to the best of our knowledge no studies have been carried out to evaluate the effect

of the resulting organic material after their addition to soil.

In addition, many recent studies explored the potential benefits of biochar as soil amendment due to

its physical and chemical properties, (Chan *et al*, 2007, Kookana *et al*, 2011). Biochar results from

the incomplete combustion or pyrolysis of various feedstock materials. The biochar production

process transforms OM into aromatic products, which are resistant against microbial decomposition

and show increased adsorption properties compared to untransformed organic matter (Lehman *et al*,

2006). As a result, the use of biochar as co-composting agent leads to a reduction of C emissions

due to adsorption of organic constituents on the biochar surface (Rogovska *et al,* 2011; Jindo *et al,*

2012, Vu *et al*, 2015).

To further enhance the protection of OM through the formation of organo-mineral or OM-biochar

associations during co-composting, the addition of worms may be a promising avenue. In general,

organo-mineral associations are enhanced by the presence of worms, due to the simultaneous

ingestion of organic matter and minerals (Shipitalo and Protz, 1989). Micro-aggregates formed

inside the worm guts improve physical protection of C (Bossuyt *et al*, 2005). However, these

interactions have only been evidenced for soil earthworms and have never been evaluated as a

strategy to reduce $CO_2$ emissions during co-composting. Moreover, to the best of our knowledge, no

studies have investigated the effect of biochar as a co-composting agent during vermicomposting.

The objective of this study was to evaluate if the addition of clay, biochar and their mixture to pre-

composted wastes can reduce $CO_2$ emissions during (vermi)composting and after use of the final

products as soil amendments. We hypothesized that C stabilization may be increased by addition of

(a) 2:1 clay such as montmorillonite, able to form organo-mineral associations; (b) biochar, able to

protect organic matter by adsorption and (c) their mixture, which could create synergistic effects. In

addition, we tested the effect of two different amounts of clay on the reduction of $CO_2$ emissions

during co-composting with and without worms and after addition to soil.

## 2. Materials and methods

### 2.1 Compost, additives and worms

A pre-composted green waste was sampled in its maturation phase at BioYvelines service, a

platform of green waste composting located 30 km West from Paris (France). The green wastes

were a mix of shredded leaves, brushwood and grass cuttings collected from households or firms

near the platform. Briefly, the composting process was performed in windrows. Aerobic conditions

and optimal humidity (approximately 45 %) were maintained through mechanical aeration and

water sprinkling. The pre-composted material was sampled after 4 months, at the beginning of the

maturation phase. Compost pH was 8.5 and the C:N ratio was 13.6 with 205.1 $mg.g^{-1}$ of organic

carbon and 13.3 $mg.g^{-1}$ of nitrogen. After sampling, the compost was air-dried and sieved at 3 mm

for homogenization.

The clay used was a 2:1 clay, purchased from Sigma-Aldrich. The clay's pH was between 2.5-3.5

and its specific surface area (SSA) was 250 m²/g. Montmorillonite was chosen because organo-

mineral interactions depend on clay mineralogy (1:1 clay or 2:1 clay). In general, 2:1 minerals offer

a bigger contact area for OM bonding and create stronger bonds with OM than the 1:1 minerals

(Kleber *et al*, 2015). Thus numerous organo-mineral associations were expected due to this large

SSA.

The biochar used was the product of gasification at 1200°C of a conifer feedstock and it was

provided by Advanced Gasification Technology (Italy). Biochar had a pH of 9.3 and a C:N ratio of

4030, with 806 mg $g^{-1}$ of organic carbon and 0.2 mg $g^{-1}$ of nitrogen (Wiedner *et al*, 2013).

*Eisenia andrei* and *Eisenia foetida* worms were purchased from La Ferme du Moutta, a worm farm

in France. The two species were chosen because they present a high rate of consumption, digestion

and assimilation of OM, can adapt to a wide range of environmental factors, have short life cycles,

high reproductive rates and endurance and resistance to handling (Dominguez and Edwards, 2011).

**2.3 Experimental setup**

The present study was designed to evaluate and compare the $CO_2$ emissions of the different organic

materials during the production phase and after their addition to soil (Fig.1)

**First step**

Co-(vermi)composting was carried out at ambient temperature in the laboratory with 10 treatments

and four replicates per treatment: (i) compost alone, (ii) compost with 25% (w/w) of

montmorillonite (low clay treatment), (iii) compost with 50 % (w/w) of montmorillonite (high clay

treatment), (iv) compost with 10% (w/w) of conifer biochar and (v) compost with a mixture of

biochar (10% w/w) and montmorillonite (25% w/w). All treatments were established with and

without worms (Table 1). Considering that a clay can retain 1 mg C per m² (Feng *et al*, 2011), 50%

of clay and 25% of clay were chosen in order to theoretically retain 60% and 30% of the total

carbon from the compost. In addition, biochar was moistened before addition to compost to avoid

worm mortality due to desiccation (Li *et al*, 2011). The addition of 10% of biochar was chosen

according to Weyers and Spokas (2011) to avoid negative effects on worms.

Worms were grown in the same compost as used in the experiment. Eight adult worms were chosen

and cleaned to remove adhering soil/compost before estimating their body mass and added to the

organic material.

The experiments were carried out in 2L jars. A dry mass of 75 g of pre-composted material was

used in each treatment. Water was sprinkled on jars at the beginning of the experiment to reach an

optimal moisture level of 80-90% (water content by weight), which was maintained throughout the

experimental period. Jars were placed in the dark at ambient temperature (24°C on average). The

(vermi)composting was stopped after 21 days, when all the organic matter should have been

ingested (75 g of compost for 8 worms). Indeed a worm can ingest its weight at maximum per day

(0.5g).

At the end of the experiment, worms were counted and weighted again. The amount of coccons and

juvenils was recorded. The final (vermi)composts were airdried, sieved at 2 mm and an aliquot was

ground for further analyses.


**Second step**

A loamy cambisol soil was collected for the laboratory experiment from the experimental site of a

long-term observatory for environmental research (ORE-ACBB) of INRA, near Lusignan in the

South-West of France. This soil was used for crop production for the last three years. The soil was

collected at depth 0-10 cm, sieved at 4 mm, homogenized and kept at 4°C until the beginning of the

experiment. The soil is carbonate-free and has the following characteristics: pH 6.4, nitrogen 1.15

$mgN\ g^{-1}$, carbon 10.56 $mgC\ g^{-1}$, sand 11%, clay 17% and silt 72% (Chabbi *et al,* 2009).

For all the treatments, 57 g of dry soil were weighed and placed into 2L glass jars. The mixtures

were homogenized through thorough mixing. All ten organic materials obtained during the first step

were applied to soil at a rate of 67g $kg^{-1}$ (dry weight). Amended and unamended soils were

incubated in four replicates in the dark at ambient temperature. Soil moisture was adjusted to 18 %

(dry weight) and maintained throughout the experiment by compensating weight losses with

deionised water. The $CO_2$ emissions were measured during 30 days as described below.

**2.3 Carbon mineralisation**

$CO_2$ emissions were measured in the headspace of the jars according to Anderson (1982). All

incubation jars contained a vial with 30 mL of 1M NaOH (first step) or 0.5M (second step) to trap

$CO_2$. The NaOH vials were covered with a tissue to avoid contamination of the NaOH solution by

worms. During the first (vermi)composting step, NaOH traps were replaced at day 1, 2, 3, 4, 8, 11,

14, 16, 18 and 21. During the incubation with soil, vials were replaced at day 1, 2, 4, 7, 14 and 22.




Phenolphalein and $BaCl_2$ solution in excess were added to a 10 mL aliquot of NaOH sampled from each vial. The solution was titrated with 1M HCl until neutrality to determine the $CO_2$-C released. Three empty jars were used as control.

Results are expressed in mg $CO_2$-C/ g compost (dry weight) or in mg $CO_2$-C/ g total organic carbon

(TOC) according to the formula:

$$Released\ CO2-C\ = \frac{(B-V)*N*E}{P}$$

where B is the volume of HCl used to titrate the control (mL); V the volume of HCl used to titrate the sample (mL); N the normality of HCL (1M); E (22) the molar mass of $CO_2$ divided by 2 (because 2 mol of $OH^-$ are consumed by one mol of $CO_2$) and P the weight of the sample (grams).


**2.4 Properties of the final products after co-(vermi)composting**

Organic carbon and nitrogen contents were measured using a CHN auto-analyzer (CHN NA 1500, Carlo Erba). A glass electrode (HANNA instruments) was used to measure pH in water extracts of (vermi)-composts (1:5). Dissolved organic carbon (DOC) contents were determined in 0.034 mol $L^-$

$^1$ $K_2SO_4$ extracts (1:5 w/v) using a total organic carbon analyzer (TOC 5050A, Shimadzu).

**2.5 Calculations and Statistical analysis**

The amount of $CO_2$-C mineralized was expressed as mgC per g of TOC, including for step 1, (vermi)compost C and biochar C and for step 2 soil C, (vermi)compost C and biochar C. Finally, a

global carbon balance was done and calculated on the basis of the $CO_2$ emissions from the composting phase and the soil incubation after amendment. These results are expressed as mgC per g of TOC, including soil C, (vermi)compost C and biochar C.

Additionally, for the composting (step 1), the amount of $CO_2$-C mineralized was expressed as mgC per g of compost in order to focus on the carbon from the pre-composted material (the amount of

biochar and clay was not included). Biochar is not supposed to be mineralized during this step



because it is produced at high temperatures and therefore its C is supposed to have a high chemical

recalcitrance against biological decomposition (McBeath and Smernik, 2009).

A first-order model was applied to describe the rate of C mineralization during composting (step 1):

$$C = C_0 (1 - e^{(-kt)}), \qquad \textit{equation 1}$$

where C is the cumulative amount of $CO_2$-C mineralized after time t (mgC $g^{-1}$ compost), $C_0$ is the

initial amount of organic carbon (mgC $g^{-1}$ compost), t is the incubation time (days), and k is the rate

constant of $CO_2$-C mineralization (day$^{-1}$).

All reported data are the arithmetic means of four replicates. A Kruskal-Wallis test was performed

to assess the significance of differences of $CO_2$ emissions from the different treatments. A Student t

test was run to investigate the influence of the different substrates on the worm development.

Significance was declared at the 0.05 level. Statistical analyses were carried out using the R 3.12

statistical package for Windows (http://www.r-project.org).

### 3.Results

**3.1 Properties of the co-(vermi)composts**

Total N, OC, DOC and pH of initial pre-composted material and of the different co-

(vermi)composts are shown in Table 2. The pH of composts and vermicomposts ranged from 7.9 to

8.7. The lowest pH was observed for the high clay treatments and the highest pH was recorded for

control treatments and (vermi)composts with biochar. Compared to the initial pH of pre-composted

organic material (8.5 ± 0.1), the high clay treatment led to a significant decrease of pH. Presence of

worms during the composting phase had no effect on pH.

Total OC in composts and vermicomposts ranged from 118.6 mg $g^{-1}$ to 241.9 mg $g^{-1}$ and total N

from 8.5 mg $g^{-1}$ to 13.5 mg $g^{-1}$. Compared to initial pre-composted material, OC was decreased

significantly after 21 days of (vermi)composting in both control treatments while N concentrations

remained unchanged. Addition of clay produced lower OC and N concentrations due to dilution,

whereas the addition of the C-containing biochar increased OC concentrations and decreased N

concentrations. Similarly to pH, the presence of worms had no effect on OC or N of co-vermicomposted material.

DOC in composts and vermicomposts ranged from 15.04 to 29.08 mg g$^{-1}$ TOC. DOC was similar to

the pre-composted material after 21 days of (vermi)composting for the two controls whereas the presence of additives significantly decreased the DOC in all other treatments. The lowest DOC concentrations were recorded for compost and vermicompost produced with biochar/clay mixture. The presence of worms had only an effect on DOC for compost produced with clay, decreasing its concentration by 12% (high clay treatment) and 16% (low clay treatment).


**3.2 Worm growth and reproduction**

The number of worms and their total weight were measured before and after the experiment. The number of worms did not vary after vermicomposting (p-value > 0.07) and neither did their total weight (p-value = 0.34). Cocoons and juveniles were separated manually from the substrates and

counted at the end of the experiments. The number of cocoons and juveniles in treatments ranged from none to 4: high and low clay treatments did not differ significantly from the control treatments (p-value= 0.39). No cocoon and no juvenile were counted in the biochar treatment. Finally, in treatments with clay/biochar mixture, the number of cocoons and juveniles was significantly higher (p-value=0.003) compared to the treatment with biochar alone with an average of 3 cocoons and

one juvenile.

**3.3 Carbon mineralisation during co-(vermi)composting**

During the composting phase (Fig. 2), the presence of worms did not change the C mineralisation (mg g$^{-1}$ TOC) in treatments free of additives. In the low clay treatments, the presence of worms decreased the amount of C mineralized. In contrast, increased C mineralization was noted for the

high clay and the biochar/clay mixture (56 % and 66% increase). The cumulative $CO_2$ emissions (mg g$^{-1}$ compost) during composting and vermicomposting did not reach a plateau for any treatment

(Fig. 3 and 4), but the experimental period was limited by worms activity since worms had processed all organic material after 21 days.

Rate constants obtained with the first-order kinetic model (eq. 1) are listed in Table 2. Similarly high rate constants suggest a rapid carbon mineralization from compost or vermicompost without additives (control). In general, treatments with worms showed higher rate constants than those without, except for the low clay treatment and control treatments. With worms, the lowest rate constant was observed for the low clay treatment. Without worms lower degradation rates as

compared to the control were recorded for treatments with high clay and clay/biochar mixture. Biochar alone decreased C mineralization more in treatments without worms.

In treatments without worms, cumulative carbon emissions at the end of the experiment ranged from 6.4 to 11.9 mg $CO_2$-C $g^{-1}$ compost, whereas in treatments with worms values ranged from 7.9 to 12.0 mg $CO_2$-C $g^{-1}$ compost (Fig. 3 and 4). In both control treatments without substrate additions,

the amounts of carbon mineralized after 21 days were similar, about 12 mg $CO_2$-C $g^{-1}$ (vermi)compost. Co-(vermi)composting with clay led to a significant decrease of the carbon emissions compared to the controls. The cumulative carbon emissions were decreased by 15% in the low clay treatment without worms and by 34% in the same treatment with worms (Fig. 3 and 4). In the high clay treatment, $CO_2$ emissions were reduced by 43% without worms and by 24% when

worms were present.

**3.3 Carbon mineralisation during incubation with soil**

Carbon emissions from the soil amended with the organic materials (step 2) are shown in Figure 5. Cumulative emissions at day 30 ranged from 8.95 to 18.20 mg $g^{-1}$ TOC. Generally, the application of organic materials to soil led to a larger amount of carbon mineralized compared to the soil

without amendments. The carbon emissions were influenced by the (vermi)compost production procedure (additives and worms). The highest emissions were recorded for soil amended with (vermi)composts free of additives. Compared to soil amended with composts, vermicomposts

decreased the carbon emissions from amended soil only when produced without additives or with

low clay addition. High clay addition during co-composting with and without worms produced

organic amendments which induced similar C emissions from soil. Compost produced in the

presence of biochar showed the lowest effect on mineralization in soil. When biochar was mixed

with clay, the co-vermicompost induced lower C emissions from soil compared to the co-compost.

Figure 6 shows the correlation between the amount of carbon mineralized from the amended soil

and the DOC of the respective organic material. The relationship was stronger for the soil amended

with composts compared to the soil amended with vermicomposts (Fig.6, respectively $R^2=0.67$ and

$R^2=0.07$).

**4.Discussion**

**4.1 Effect of worms and additives on compost properties**

280        After 21 days of vermicomposting, OM had been processed into a homogeneous and aerated

material whereas composts had a compact aspect, illustrating the positive effects of worms on the

physical structure of the final product. Co-composting with biochars did not lead to any change in

pH of the final product (Table 2). This may be due to the alkaline pH of the pre-composted material

and the low amount of biochar added. In contrast, the addition of acidic clay (pH 2.5 to 3.5) to

slightly alkaline pre-composted material tended to reduce the pH of the final (vermi)compost (Table

2). The presence of worms in our experiments had no effect on the pH, all the treatments tending to

a slightly alkaline pH. By contrast, some authors observed a decrease in pH during

vermicomposting of household wastes (Frederickson *et al*, 2007) or cattle manure (Lazcano *et al*,

2008). The contrasting results may be explained by a lower production of $CO_2$ and organic acids by

micro-organisms in our experiment due to the almost mature pre-composted material used

compared to the fresh green wastes used in previous experiments.

        The C:N ratio was significantly higher in (vermi)composts produced in the presence of

biochar, due to addition of carbon enriched material. No difference was observed in vermicompost

treatments compared to compost treatments, concerning the OC and the total N. These results are in

line with those obtained by Ngo *et al* (2013), who suggested that the elemental composition and the

chemical structures present in different composts and vermicomposts could be similar.

**4.2 Effect of worms and additives on carbon mineralization during (vermi)composting (step1)**

Data recorded (Fig. 3 and 4) for control treatments indicated in contrast to what is generally

observed (e.g. Chan *et al*, 2011), that the presence of worms did not lead to higher $CO_2$ emissions

during composting. This is probably due to the OM used, which was almost mature compost and,

may thus be characterised by lower degradability than the organic wastes originating from

households usually used for (vermi)composting.

Addition of clay and biochar reduced carbon emissions during composting (Fig. 3). Similar results

were obtained by other authors for co-composting in absence of worms with clay additives (Bolan

et al, 2012) or biochar (Dias *et al*, 2010). These data may indicate carbon stabilization by physico-

chemical protection of OM on clay and/or biochar surfaces. Carbon storage generally increases

linearly with increasing clay concentration (Hassink, 1997). This is in line with our results, showing

proportional $CO_2$ decrease, when clay content and thus surface area was doubled.

By contrast, in the presence of worms, C mineralization was more reduced for the low clay

compared to the high clay treatment (Fig. 4). As we observed similar worm biomass in both

treatments, we hypothesize that high clay contents may have negative effects on worm activity and

therefore the formation of organo-mineral associations. This hypothesis is supported by the results

of Klok *et al* (2007), who showed that *Lumbricus rubellus* worms can have their life cycle

influenced by a high content of clay in soil leading to anaerobic conditions and soil compaction.

Our results suggest that a 50% proportion of montmorillonite also impacts the activity of *Eisenia*

species. In contrast, in the low clay treatment, worm activity most probably increases the formation

of organo-mineral associations, thus leading to higher reduction of $CO_2$ emissions compared to

regular composting without worms (Fig. 3). These results indicate that the protective capacity of



clay minerals may be enhanced by worm activity, up to a threshold of the clay:OM ratio, above

which *Eisenia* species are no longer able to reduce $CO_2$ emissions. *Eisenia* species belong to the

epigenic worm species living at the soil surface in leaf litter, one of the three ecological lifetypes

described by Bouché (1977). Therefore, they are well adapted to process pure organic matter and

may be less suited for co-composting with minerals. The optimal clay: OM ratio to allow for

maximal reduction of $CO_2$ emissions remains to be assessed as well as the possibility to use other

worm species more adapted to ingestion of minerals.

When biochars were added, alone or in mixture with clay, contrasting results were observed among

co-composting processes with and without worms. In general biochar addition led to a reduction of

$CO_2$ emissions up to 44% compared to the regular (vermi)compost (Fig. 2 and 3). In absence of

worms, a 24% decrease was observed in treatments with biochar and a 46 % decrease with

biochar/clay mixture (Fig. 2 and 3). These results are in contrast with other studies showing no

significant reduction of $CO_2$ emissions when biochar was used for co-composting (Sánchez-García

*et al*, 2015). However, biochar effects may depend on its physic-chemical properties, which are

depending on the production conditions. Therefore, the biochar produced by gasification used in

this study may have different effects compared to biochar produced by pyrolysis used in the other

study. Reduced $CO_2$ emissions in presence of biochar are in line with observation by other authors

concerning biochar effects on microbial activity and OM mineralisation. Decrease of OM

mineralization induced by biochar was explained by its capacity to adsorb labile organic

compounds, which may otherwise be degraded (Augustenborg *et al*, 2012; Ngo *et al*., 2013; Naisse

*et al*., 2015).

In the presence of worms, the addition of biochar and biochar/clay mixture induced higher $CO_2$

emission (p-value > 0.1) compared to regular co-composting (Fig. 2 and 3). Three hypothesis might

explain that worms drastically modify the complex interactions between clay, biochar and pre-

composted OM: 1) the worms might increase in their gut the contact between clay and biochar,




leading to the partial saturation of clay surfaces with carbon compounds originating from biochar and thus to a reduction of the available surface area; 2) the microbial colonization of biochar might be enhanced in the worm gut decreasing their long-term resistance to bio-degradation; 3) biochars might enhance worm activity, as suggested by Augustenborg *et al* (2012) to explain the increase of

$CO_2$ emissions when biochar was added to soil in the presence of worms.

The incidence of these three hypotheses probably depends on the biochar quality, which influences the affinity of worms for biochar. The properties of the biochar and its effects on the worms might be dependent upon their production process, for example upon the initial feedstock or the conditions of pyrolysis. In our case, biochar had no effect on worm biomass, but it had contradictory effects on

worm reproduction. The addition of biochar alone reduced the number of juveniles and cocoons of *Eisenia* to zero, indicating a high stress for worms. For the development of soil earthworms, the presence of biochar has already been described as a potential risk (Liesch *et al*, 2010). In soil, the negative effects of biochar on worm activity have been suggested to originate from a) a lack of nutrients following their adsorption on biochar, b) the presence of toxic compounds such as

polycyclic aromatic hydrocarbons (PAH) mainly, or c) a lack of water (Li *et al*, 2011). In our experiment, the lack of nutrients was balanced by the presence of compost and the lack of water was avoided by a preliminary humidification of biochar before their addition. The presence of PAH or other potentially toxic substances might thus explain the negative effects that we observed. Further analyses and longer experiments should be carried out in order to investigate the reasons for

these stressful effects and for the increasing reproduction rates observed when biochar was used in combination with clay. Testing the influence of biochar with contrasted origins (initial material and process) on vermicomposting with clay compared with similar composting treatments would be necessary to elucidate the mechanisms responsible for their influence on C mineralization.

**4.3 Effect co-(vermi)compost production conditions on carbon mineralization in soil (step 2) and total carbon balance**





The production conditions had more influence on C mineralization in soil for composts than vermicomposts (Table 3). Clay and biochar reduced the concentration of labile compounds in composts and vermicomposts leading to decreased DOC concentrations of the final amendments

(Table 1). But the $CO_2$ emissions after addition to soil were only reduced by clay and biochar addition when the compost was produced without worms (Fig 5). The rate of mineralization of organic amendments is generally linked to the labile carbon compounds, (Chaoui *et al*, 2003) as was observed for the compost addition. The lack of correlation between DOC and $CO_2$ emitted after addition to soil of vermicomposts suggests contrasted properties of DOC in composts and

vermicomposts (Lazcano *et al*, 2008, Kalbitz *et al*, 2003).

$CO_2$ emitted from soil after the addition of amendments may originate from two sources: the mineralization of added carbon and the mineralization of native soil OM. Differences compared to the control (soil without amendments) may be explained by positive or negative priming effect, induced by microbial reaction to OM addition.

In case of amendments produced with biochar alone, a negative priming effect could be observed, because the mineralization rate observed for this treatment was lower than for the control. This result is in line with many other studies reporting reduced mineralization of native soil OM after biochar amendment (Zimmerman *et al*, 2011). Our data evidenced that this phenomenon may also occur for composts, when biochar is used as co-composting agent. This was not observed for co-

composts produced in the presence of worms. However, the addition of worms may attenuate these effects of the final product.

In order to evaluate the positive or negative effect of each additive and process (with or without worms) on C mineralization, the emitted $CO_2$ of both steps (composting phase and incubation of soil with amendments) was summed up and expressed as mgC $g^{-1}$ TOC. The carbon emissions

during step 1 and step 2 were influenced differently by the (vermi)compost production procedure (additives and worms). The lowest total carbon emissions were recorded for compost and vermicompost produced in presence of biochar. Low clay treatment more efficiently reduced carbon



emissions in the presence of worms. In all other treatments, except the control, $CO_2$ emissions were higher for vermicompost due to higher emissions during the production step.


## 5. Conclusion

This study tested the possibility to use the worm species *Eisenia* to reduce $CO_2$ emissions during co-composting of green wastes with clay, biochar or their mixture. We established the complete carbon balance taking into account production of amendments as well as the effect after their

addition to soil. Most additives were found to decrease $CO_2$ release during co-(vermi)composting. In the presence of worms, 25% of clay led to greater OM protection than 50%. The opposite was observed in the absence of worms. Our results thus evidenced a threshold of clay concentrations for *Eisenia* worms, above which $CO_2$ emissions are no longer reduced. Biochar had a positive effect on carbon storage for all treatments. Biochar/clay mixture resulted in synergistic effects for treatment

without worms. We conclude that the use of additives may have the potential to greatly reduce $CO_2$ emissions during co-composting. Worms further reduced $CO_2$ emissions only in treatment with low clay dose. The effect of the amendments on C mineralization after addition to soil was small in the short-term. We suggest that production conditions during co-(vermi)composting have to be optimized in terms of total $CO_2$ reduction by choosing the minerals, their optimal ratio with OM

and testing different worm species. The effects of these amendments on soil fertility and plant growth remain to be investigated. Further work need to be done to assess the long-term effect of these amendments.

### 6.Acknowledgements

Authors are grateful to the Pierre and Marie Curie University for Doctoral fellowship. We also acknowledge Daniel Billiou and Valérie Pouteau for their technical assistance. Financial support was provided by CNRS under the framework EC2CO program (LOMBRICOM project), ADEME under the framework of the DOSTE program (VERMISOL project) and by the Australian Research





Council by a discovery grant (C corundum project).





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






**Table 1: Mean values of pH, DOC, content of total nitrogen and organic carbon after 21 days of co-(vermi)composting.**

| | pH | C (mg g$^{-1}$) | N (mg g$^{-1}$) | DOC (mg g$^{-1}$ TOC) | C/N |
|---|---|---|---|---|---|
| Pre-composted material | 8.5 ± 0.1 [c] | 205.1 ± 3.0 [b] | 13.3 ± 0.2 [a] | 29.08 ± 0.86 [a] | 15.4 ± 0.1 [bc] |
| **Organic materials after 21 days of co-composting** | | | | | |
| **Compost treatments** | | | | | |
| C | 8.7 ± 0.1 [ab] | 188.2 ± 9.1 [c] | 13.5 ± 0.8 [a] | 28.85 ± 0.38 [a] | 13.5 ± 0.6 [d] |
| C + 25 % M | 8.2 ± 0.1 [d] | 153.1 ± 9.5 [d] | 10.6 ± 0.5 [c] | 21.77 ± 1.57 [b] | 14.4 ± 0.8 [d] |
| C + 50 % M | 7.9 ± 0.1 e | 118.6 ± 2.9 [e] | 8.5 ± 0.1 [e] | 19.32 ± 0.94 [c] | 14.0 ± 0.3 [d] |
| C + 10 % B | 8.7 ± 0.1 [a] | 241.9 ± 15.1 [a] | 12.4 ± 0.5 [b] | 21.26 ± 0.78 [b] | 19.5 ± 0.8 [ab] |
| C + 10 % B+ 25 % M | 8.2 ± 0.1 [d] | 197.8 ± 5.9 [b] | 10.0 ± 0.2 [cd] | 15.04 ± 0.68 [e] | 19.7 ± 0.3 [a] |
| **Vermicompost treatments** | | | | | |
| V | 8.6 ± 0.1 [b] | 185.0 ± 8.3 [c] | 13.0 ± 0.6 [ab] | 26.83 ± 0.49 [a] | 14.3 ± 0.4 [d] |
| V + 25 % M | 8.2 ± 0.1 [d] | 150.2 ± 5.2 [d] | 10.4 ± 0.5 [cd] | 18.41 ± 0.66 [cd] | 14.5 ± 0.3 [d] |
| V + 50 % M | 7.9 ± 0.1 [e] | 121.4 ± 6.0 [e] | 8.6 ± 0.1 [e] | 17.16 ± 0.7 [d] | 14.1 ± 0.7 [d] |
| V + 10 % B | 8.7 ± 0.1 [ab] | 247.6 ± 12.3 [a] | 12.5 ± 0.5 [b] | 19.68 ± 0.49 [bc] | 19.9 ± 0.9 [a] |
| V + 10 % B+ 25 % M | 8.3 ± 0.1 [d] | 206.0 ± 11.4 [b] | 9.9 ± 0.3 [d] | 15.18 ± 0.43 [e] | 20.8 ± 1.4 [a] |

**Table 2: Effect of the addition of clay and/ or biochar on the rate constant k (day$^{-1}$) during composting and vermicomposting.**

| | k (10$^{-3}$ day$^{-1}$) | Std. Error (10$^{-5}$) |
|---|---|---|
| **Compost treatments** | | |
| C | 3.069 [a] | 4.429 |
| C + 25 % M | 2.588 [cd] | 4.539 |
| C + 50 % M | 1.699 [g] | 2.776 |
| C + 10 % B | 2.313 [ef] | 2.204 |
| C + 10 % B+ 25 % M | 1.762 [g] | 5.265 |
| **Vermicompost treatments** | | |
| V | 3.036 [ab] | 4.089 |
| V + 25 % M | 1.973 [fg] | 3.783 |
| V + 50 % M | 2.431 [de] | 3.616 |
| V + 10 % B | 2.855 [ab] | 4.869 |
| V + 10 % B+ 25 % M | 2.798 [bc] | 4.251 |





**Table 3: Carbon balance.** Data are presented as means and standard error (n=4). Different small letters indicate significant differences between treatments (Kruskal-Wallis test, $p < 0.005$)

| | Composting phase (mgC g$^{-1}$ TOC) | Soil incubation phase (mgC g$^{-1}$ TOC) | Total carbon mineralized (mgC g$^{-1}$ TOC) |
|---|---|---|---|
| **Compost treatments** | | | |
| C | 17.11 [a] | 18.20 [a] | 35.31 [a] |
| C + 25 % M | 13.55 [b] | 15.68 [ab] | 29.23 [a] |
| C + 50 % M | 7.83 [bc] | 14.03 [bc] | 21.87 [de] |
| C + 10 % B | 8.67 [de] | 8.95 [f] | 17.62 [f] |
| C + 10 % B+ 25 % M | 6.36 [e] | 13.58 [c] | 19.94 [ef] |
| | | | |
| **Vermicompost treatments** | | | |
| V | 15.75 [a] | 13.11 [cd] | 28.87 [ab] |
| V + 25 % M | 10.59 [c] | 13.72 [c] | 24.31 [cd] |
| V + 50 % M | 12.23 [bc] | 13.73 [c] | 25.96 [bc] |
| V + 10 % B | 8.81 [d] | 11.42 [ef] | 20.22 [ef] |
| V + 10 % B+ 25 % M | 10.59 [c] | 12.67 [de] | 23.27 [cd] |




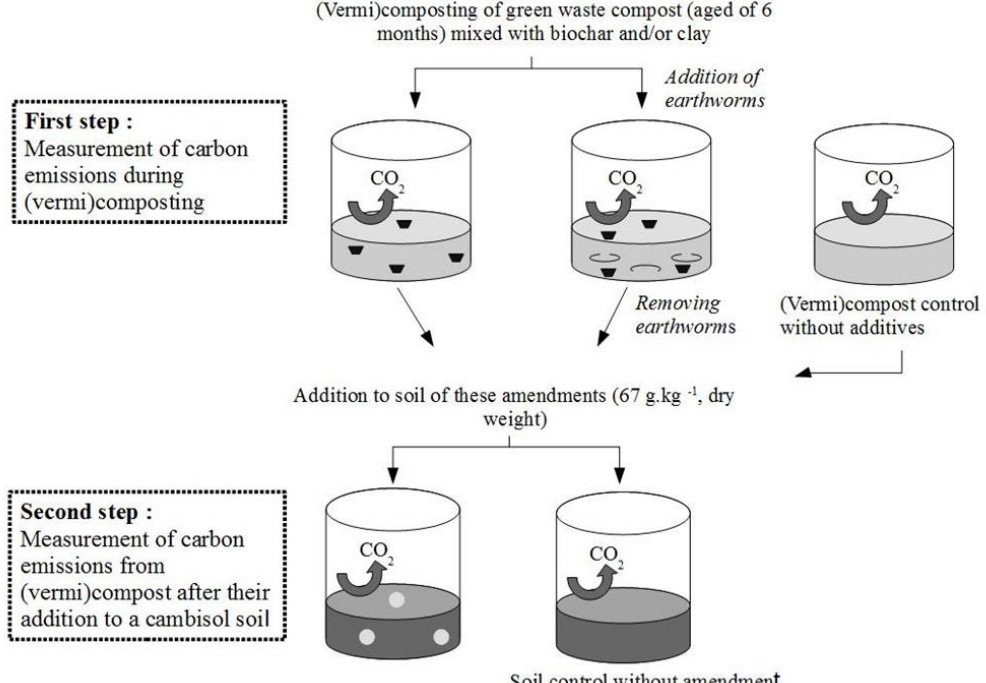

Figure 1. Experimental design to compare $CO_2$ emissions of different organic materials during composting and after their addition to soil.

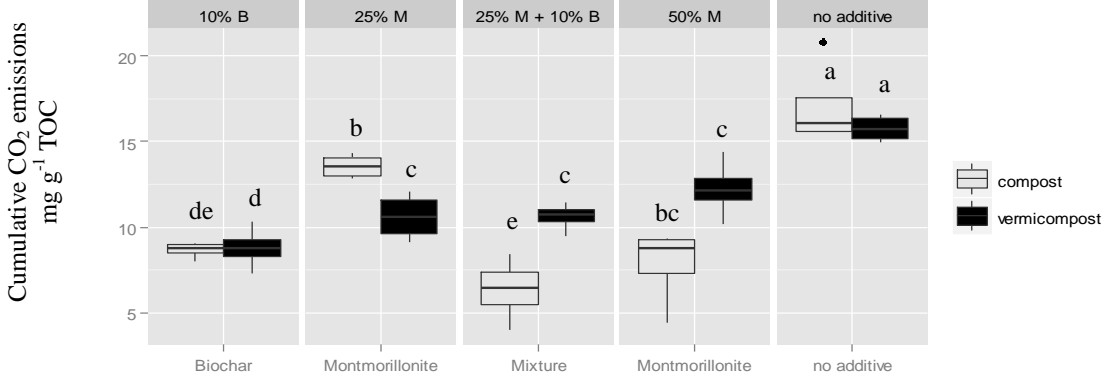

Figure 2. Cumulative $CO_2$ emissions at day 21 from composts and vermicomposts. Letters a,b,c, d, e and f means the statistical difference.



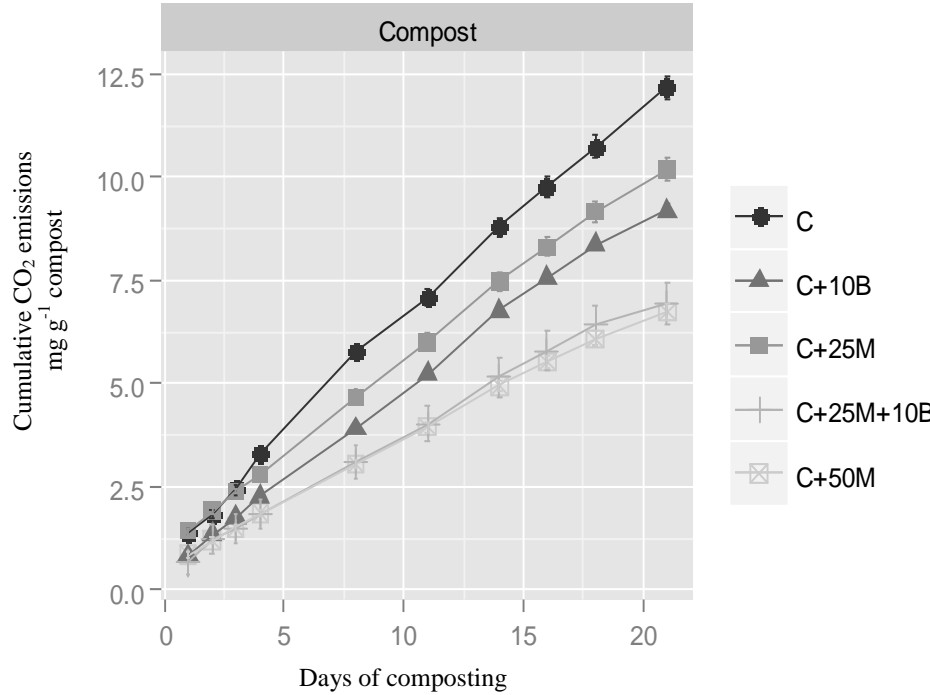


**Figure 3: Cumulative CO₂ emissions during composting without worms of pre-composted material alone (C), with 25% of clay (C+25% M), with 50% of clay (C+ 50% M), with 10% of biochar (C+ 10%B) and, with 25% of clay and 10% of biochar (C+25%M + 10% B).**





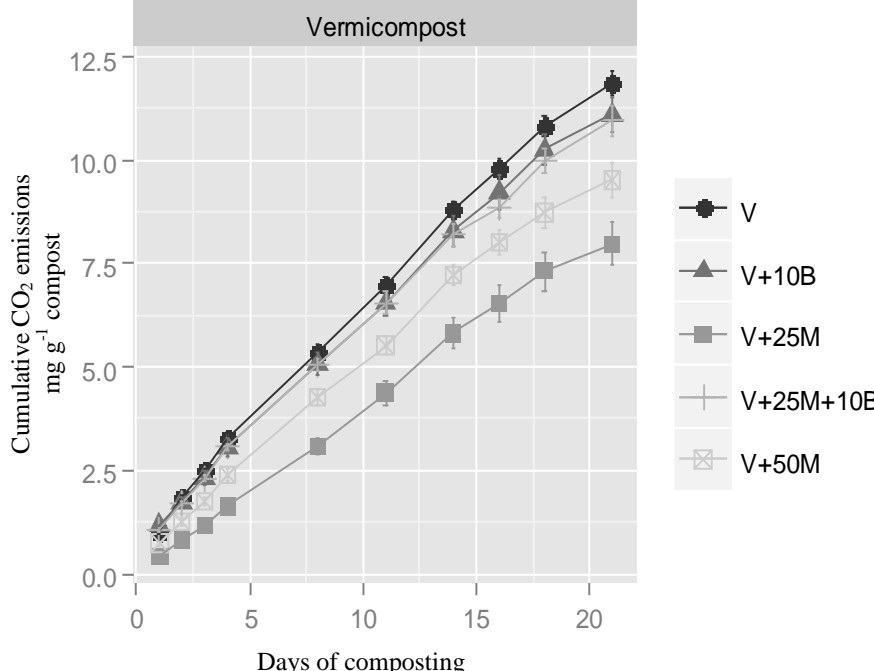

**Figure 4: Cumulative CO₂ emissions during composting with worms of pre-composted material alone (V), with 25% of clay (V+25% M), with 50% of clay (V+ 50% M), with 10% of biochar (V+ 10%B) and, with 25% of clay and 10% of biochar (V+25%M + 10% B).**





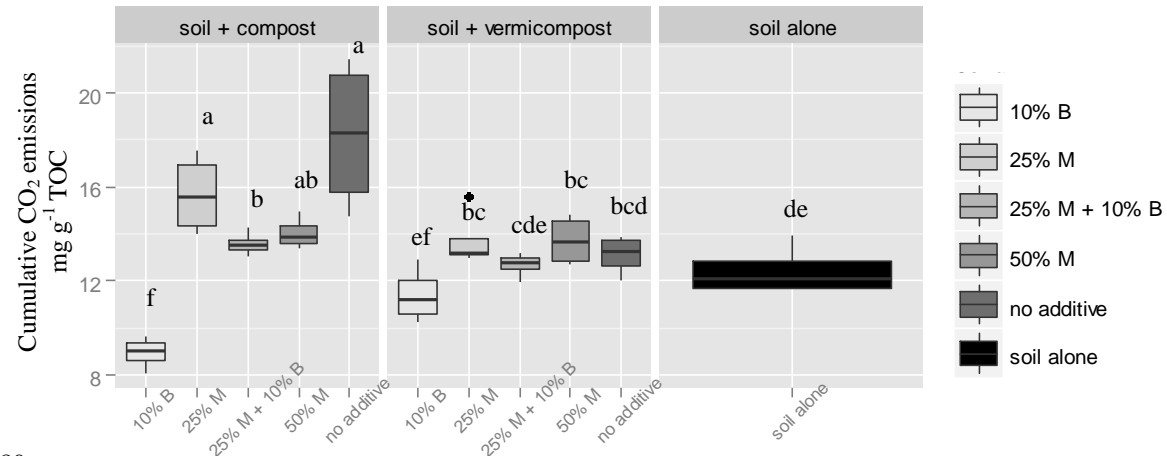


**Figure 5. Cumulative CO$_2$ emissions at day 30 from composts and vermicomposts in soil.**

Letters a,b,c, d, e and f means the statistical difference.





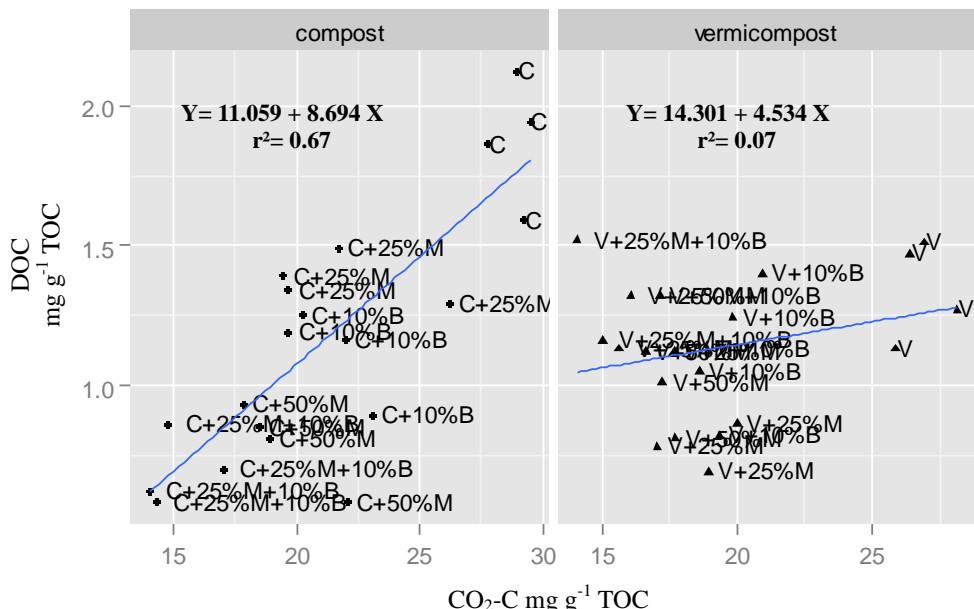

**Figure 6. Comparison between cumulative $CO_2$ emissions at day 30 from composts and vermicomposts in soil and DOC from these amendments.**
