# Peer review of "The effects of worms, clay and biochar on CO2 emissions during production and soil application of co-composts"

_SOIL, 2016_

## Referee Comment (RC1) · Anonymous Referee #1 · 28 Jun 2016

This interesting study has as its goal to determine if adding clay, biochar, or both to precomposted wastes can reduce CO2 emissions during compositing or vermicomposting, and "after the use of the final products as soil amendments." The experiments seem well-designed, and the paper's results are useful.

I recommend two major revisions to this paper focusing on 1) logical flow and 2) figure quality and clarity.

1. logical flow: The paper struggles with logical flow in a few places, and the paper would be stronger if this were addressed prior to publication. Parts of the paper seem

to be advancing the argument that clay, biochar, and worms can act together to reduce $CO_2$, while in other places the paper stays closer to the data, which mostly do not show this. The paper would still be interesting if the authors argued (as the data seem to) that most combinations of biochar, clay, and worms speed up soil $CO_2$ loss. However, parts of the text are confusing because of the advocacy for adding worms to reduce $CO_2$ emissions. Figure 2, which shows $CO_2$ emissions vs treatment, does not show that worms help reduce $CO_2$ emissions except in one of the five scenarios, with the other 4 scenarios leading to either increased $CO_2$ emissions or unchanged $CO_2$ emissions. This makes sentences like this one, from the end of the abstract (line 28), seem like a non sequitur: "In summary, the addition of worms during co-composting with clay and biochar may be a promising technology for reducing $CO_2$ emissions and increasing soil carbon storage."

The paper's title would benefit from revision for clarity. The use of a question as a title suggests that the authors have not yet decided what the conclusions of the work are. Either a conclusive title (e.g. "Worms can increase $CO_2$ emissions during co-composting with clay and biochar") or a declarative title (e.g. "The effects of worms on $CO_2$ emissions during co-composting with clay and biochar") would strengthen the paper.

Section 3.3 would benefit from focusing and shortening.

Section 4.1 should be moved to the results section.

Section 4.2 would also benefit from focusing and shortening. For example, the paragraph starting at line 328 ("When biochars were added. . .") would benefit from a thesis statement – what is the main idea that this paragraph is conveying? The paragraph starting on line 351 would also benefit from similar focusing.

2) Figure quality and clarity: Figure 2 captions: suggest revising "Letters a,b,c, d, e, and f means the statistical difference" to "columns with the same letter are statistically identical," or "different letters indicate statistically significant differences."

Figures 3, 4, and 5: legibility would be improved if the axes numbers were in black font, not gray

Figure 6: this figure is not comprehensible because the text is so highly overprinted.

Other items: Introduction: Paragraph beginning at line 37: this paragraph would benefit from proofreading for clarity. For example, the second sentence does not logically follow from the first. In addition, this paragraph would benefit from proofreading for punctuation (line 45).

Line 127: are the units here of mg/m correct? Should they be mg/mg instead? Lines 142-143: typos: "weighted" should be "weighed," "juvenils" -> "juveniles," and "airdried" -> "air dried."

Line 190: the assumption that biochar does not release any $CO_2$ during incubation experiments is questionable, as there are many studies showing release of $CO_2$ during biochar incubation. However, it's possible that the authors have found an outlier condition where this is the case, by choosing a biochar made at extremely high temperatures (1200°C). Please make this clear.

Line 309: provide data supporting the statement that the soil surface area doubled.

Line 402: change "to use" to "of using";

Throughout the text: Eisenia is a genus, not a species.

---

## Referee Comment (RC2) · Anonymous Referee #2 · 11 Jul 2016

Comments: 1. line 83-84 - the authors say that there are no studies that investigated the effect of biochar as a co-composting agent during vermicomposting, I would point out the research reported by Malińska et al. 2016 in Ecological Engineering on "The effect of precomposted sewage sludge mixture amended with biochar on the growth and reproduction of Eisenia fetida during laboratory vermicomposting" - the authors investigated the effect of biochar on reprodcution rate, biomass and mortality, 2. biochar was produced in unusually high temperature, what is the reason for that? 3. did the authors analyze the chemical composition of biochar? 4. also, did the authors analyze biochar for surface area and microporosity? 5. there is little information about com-

posting in the article, i.e. the duration of composting (from my udnerstanding that was the first step), the size of the reactors, temperature during composting.

---

## Author Comment (AC1) · 15 Jul 2016

Dear reviewer 1,

We sincerely appreciate your interest in our study and would like to thank you for your time and effort. We took into consideration your recommendations and suggestions you've made to improve the paper. The following was done:

Reviewer 1: This interesting study has as its goal to determine if adding clay, biochar, or both to precomposted wastes can reduce CO2 emissions during compositing or vermicomposting, and "after the use of the final products as soil amendments." The

experiments seem well-designed, and the paper's results are useful.

Thank you very much.

I recommend two major revisions to this paper focusing on 1) logical flow and 2) figure quality and clarity.

Ok, we revised the paper in particular with regards to these comments.

1. logical flow: The paper struggles with logical flow in a few places, and the paperwould be stronger if this were addressed prior to publication. Parts of the paper seemto be advancing the argument that clay, biochar, and worms can act together to reduce $CO_2$, while in other places the paper stays closer to the data, which mostly do not show this. The paper would still be interesting if the authors argued (as the data seem to) that most combinations of biochar, clay, and worms speed up soil $CO_2$ loss. However, parts of the text are confusing because of the advocacy for adding worms to reduce $CO_2$ emissions. Figure 2, which shows $CO_2$ emissions vs treatment, does not show that worms help reduce $CO_2$ emissions except in one of the five scenarios, with the other 4 scenarios leading to either increased $CO_2$ emissions or unchanged $CO_2$ emissions. This makes sentences like this one, from the end of the abstract (line 28), seem like a non sequitur: "In summary, the addition of worms during co-composting with clay and biochar may be a promising technology for reducing $CO_2$ emissions and increasing soil carbon storage."

We thouroughly revised the text in order to stay more closely to the data and removed inadequate statements advocating worms' beneficial effect on reduction of $CO_2$ emissions. In particular:

Abstract: Consequently, the sentence (line 28) was changed to "In summary, the addition of worms during co-composting with clay and biochar speeds up $CO_2$ emissions in most cases. Therefore, the production of a low $CO_2$ emissions..."

In the objectives (line 85), "their mixture to pre-composted wastes can influence $CO_2$

emissions". Word "reduce" will be changed by "influence".

Line 88: "two different amounts of on the changes of CO2 emissions". Word "reduction" will be changed to "changes".

In the conclusion: We finally started our conclusion with "This study tested the influence of worm species of the Eisenia genus on CO2...". At the end, we changed one sentence to "worms generally speed up carbon mineralization except in treatment with low clay dose".

The paper's title would benefit from revision for clarity. The use of a question as a title suggests that the authors have not yet decided what the conclusions of the work are. Either a conclusive title (e.g. "Worms can increase CO2 emissions during co-composting with clay and biochar") or a declarative title (e.g. "The effects of worms on CO2 emissions during co-composting with clay and biochar") would strengthen the paper.

Ok, the title was changed. It reads now: "The effects of worms, clay and biochar on CO2 emissions during production and soil application of co-composts".

Section 3.3 would benefit from focusing and shortening.

The results paragraph will be shortened as you suggested.

Section 4.1 should be moved to the results section.

Most of this section was moved into the section 3.1, but we prefer to keep some sentences in this section for discussing these results.

Section 4.2 would also benefit from focusing and shortening. For example, the paragraph starting at line 328 ("When biochars were added") would benefit from a thesis-statement – what is the main idea that this paragraph is conveying? The paragraph starting on line 351 would also benefit from similar focusing.

Line 328: This paragraph will focus on the treatments without worms and the potential

effect of the biochar on the carbon emissions and will be shortened.

Line 351: In the same way, this paragraph will be shortened and will focus on the biochar addition on the worm activity, potentially explaining the carbon emissions.

2) Figure quality and clarity:

Figure 2 captions: suggest revising "Letters a,b,c, d, e, and f means the statistical difference" to "columns with the same letter are statistically identical," or "different letters indicate statistically significant differences."

Suggestions to clarify and enhance the quality of the figures will be take into account. Text above the figure 2 will be changed into "Different letters indicate statistically significant differences".

Figures 3, 4, and 5: legibility would be improved if the axes numbers were in black font, not gray

Figures 3, 4 and 5 Axis number will be in black font

Figure 6: this figure is not comprehensible because the text is so highly overprinted

Figure 6: Figure was changed in order to ameliorate its comprehension (text will be moved and reduced)

Other items: Introduction: Paragraph beginning at line 37: this paragraph would benefit from proofreading for clarity. For example, the second sentence does not logically follow from the first. In addition, this paragraph would benefit from proofreading for punctuation (line 45).

Line 37: As you mentionned, the second sentence does not logically follow the first. We finally started with "Land use changes are responsible for the steady increase of CO2 in the atmosphere, along with industrial activity and the use of fossil fuels."

Line 45: Sentence were change to "Increasing soil C may be possible with the use of

composted organic wastes as alternative fertilisers (Ngo...), which could counterbalance...". Punctuation has been added.

Line 127: are the units here of mg/m correct? Should they be mg/mg instead? Lines 142-143: typos: "weighted" should be "weighed," "juvenils" -> "juveniles," and "airdried" -> "air dried."

The units mg/m$^2$ used are correct because we mentionned a specific surface area.

Line 142-143: corrections will be done: "weighed", "juveniles","air dried".

Line 190: the assumption that biochar does not release any CO2 during incubation experiments is questionable, as there are many studies showing release of CO2 during biochar incubation. However, it's possible that the authors have found an outlier condition where this is the case, by choosing a biochar made at extremely high temperatures (1200C). Please make this clear.

Line 190: We have made the assumption that the biochar used in this study does not release any CO2 during incubation experiments, based on the previous study of Naisse et al ( 2014, Effect of physical weathering on the carbon sequsetration potential of biochars and hydrochars in soil). In this study, the biochar was produced at 1200°C and in the same conditions as our biochar. Their results showed that the total soil carbon mineralized amended with biochar was under 0.5 mgC g soil-1 for 200 days. Furthermore, we will make clear our assumption by adding this sentence: "Biochar produced at high temperatures showed a very low carbon emissions during a 200 days incubation in soil (Naisse et al, 2014), so that we can hypothesize that its mineralization can be neglected compared to OM mineralization during 21 days".

Line 309: provide data supporting the statement that the soil surface area doubled.

Line 309: We could not provide data that the soil surface area doubled. Consequently, we will change the sentence by "This is in line [...], when clay content and thus potentially available surface area increased".

Line 402: change "to use" to "of using";

Ok, was done.

Throughout the text: Eisenia is a genus, not a species

Ok, we changed.

Please also note the supplement to this comment:
http://www.soil-discuss.net/soil-2016-35/soil-2016-35-AC1-supplement.pdf

---

## Author Comment (AC2) · 15 Jul 2016

Dear reviewer 2, We sincerely appreciate your interest in our study and would like to thank you for your time and effort. We took into consideration your recommendations and suggestions you've made to improve the paper. The following was done:

Reviewer 2: Comment 1. Line 83-84-the authors say that there are no studies that investigated the effect of biochar as a co-composting agent during vermicomposting, I would point out the research reported by Malinska et al.2016 in Ecological Engineering on "The effect of precomposted sewage sludge mixture amended with biochar on

the growth and reproduction of Eisenia fetida during laboratory vermicomposting"-the authors investigated the effect of biochar on reproduction rate, biomass and mortality,

Line 83-84: we will add this reference to our study and take it into account in our discussion. Thank you for this information and reference.

Comment 2: biochar was produced in unusually high temperature, what is the reason for that?

The biochar used in this study was produced at high temperature (1200°C) by gazification. Biochar obtained from the gasification system shows similar physico-chemical caracteristics as biochar obtained with pryolysis, but these biochars have a different structure, with higher macroporosity (Brewer et al, 2009). Moreover, the production at high temperature may imply a high chemical recalcitrance against biological decompostion. We used this biochar because it was well characterised in particular with regards to absence of toxic compounds like PAH and dioxine (Wiedner et al., 2013), which might have influenced worms' activity.

Comment 3: did the authors analyze the chemical composition of biochar? and 4: also, did the authors analyze biochar for surface area and microporosity? : The chemical composition of the biochar used has not been analyzed in this study. But we used the same biochar as previous authors (Wiedner et al, 2013) of agro-inductrial biomass on a commercial scaleÂăwho have analyzed its. This information was added to our manuscript. Unfortunately, we did not analysed the specific surface area and microporosity of this biochar.

Comment 5: there is little information about composting in the article, i.e the duration of composting (from my understanding that was the first step), the size of the reactors, temperature during composting.

In this study, we did not consider the whole composting phase because we used a pre-composted material from a composting plat-form in windrows. As mentionned in

the section 2.1, the pre-composted material was sampled after 4 month of composting in windrows. In our study, we used this pre-composted material mixed with different additives and ended the composting process in 2L jars. The temperature of our step was maintained to 20°C , due to the presence of worms. This information was added to the manuscript.

References

Brewer C.E, Schmidt-Rohr K., Satrio J.A. and Brown R.C. (2009) Characterization of iochar from fast pyrolysis and gasification systems. Environmental Progress and Sustainable Energy 28-3, 386-396.

Wiedner K., Rumpel C., Steiner C., Pozzi A., Maas R. and Glaser B. (2013) Chemical evaluation of chars produced by thermochemical conversion (gasification, pyrolysis and hydrothermal carbonization) of agro-industrial biomass on a commercial scale. Biomass and bioenergy 59, 264-278.

Please also note the supplement to this comment:
http://www.soil-discuss.net/soil-2016-35/soil-2016-35-AC2-supplement.pdf

---

## Author Response (AR1)

Dear Editor of Soil,

Thank your for your message. We did our best to take into account every comments and suggestions and we hope that the new version of our manuscript has been improved in term of quality.

Considering the points asked, here your find our improvements :

-In order to be clear to all readers, we have precised our objectives in the introduction and changed the subtitles « step 1 » and « step 2 » by « composting » and « soil incubation ». Moreover, the word « co-(vermi)composting » has been deleted and changed by « composting » followed by « in presence of worms » or « in absence of worms ».

- The term windrow will be defined as followed in the manuscript «  Briefly the composting process was performed in windrow which are long narrow piles of green waste ».

-Figures have been revised according to the reviews. For the figure 6, the name of each treatment has been deleted because the figure was not readable if they were written, so we have changed them by symbols, according to the treatment.

You will find the latest version, taking into account the remarks and suggestions highlighted in yellow.

Kind regards,

Justine Barthod

[revised manuscript text omitted]

---

## Author Response (AR2)

Dear Editor,

Thank you for your revision and your new suggestions to improve the manuscript.

According to your comments, we have changed the organization of the manuscript.

Now, the manuscript explains firstly the effect of clay and/or biochar on the composting process and carbon mineralization, then we have considered the effect of worm addition. Consequently, we have reorganized our hypothesis and conclusion too in this way. Some results have also been reorganized to firstly focus on the biochar/clay effect instead of the worm addition.

The discussion has been divided into several paragraphs:

*-4.1 Effect of worms and additives on compost properties*
*-4.2 Effect of additives on carbon mineralization during composting*
*-4.3 Effect of worms on carbon mineralization during composting with clay*
*-4.4 Effect of worms on carbon mineralization during composting with biochar and biochar/clay mixture*
*-4.5 Effect of co-(vermi)compost production conditions on carbon mineralization in soil and total carbon balance*

All the changes are highlighted in yellow in this latest version.

We hope that this new organization of the manuscript will meet your expectations.

Kind regards,

Justine Barthod

[revised manuscript text omitted]

---

## Author Response (AR3)

Dear Elizabeth,

Thank you for your suggestions. We think that we had already responded to the previous suggestions. However, in response to your concerns, we introduced in the introduction some studies showing that the effect of worms on soil carbon sequestration and greenhouse gas emissions is unclear. We further improved the discussion: the first paragraph on compost properties was removed from the discussion and some points were included in the results. We added a section 4.2 'effect of worms', which was divided into three subsections: 4.2.1 Effect of worms on CO2 emissions during composting without additives, 4.2.2 Effect of worms on carbon mineralization during composting with clay  and 4.2.3 Effect of worms on carbon mineralization during composting in treatments with biochar. We kept the paragraph on biochar additives separated, since their effects induce very different mechanisms compared to other additives. We also changed the conclusion in order to clarify the take-home message.

Best regards,

Justine Barthod

---

## Author Response (AR4)

Dear editor,

Thank you for your new revision and suggestions to improve this manuscript. We hope we have taken into account every comments of the reviewer in this new version. According to these comments, we have changed the structure of the discussion, now it is divided into 3 parts and the headers of each part have been changed also (*4.1 Effect of clay and biochar on carbon mineralization during composting; 4.2 The presence of worms modifies unexpectedly the effect of clay and biochar on $CO_2$ emissions during composting; 4.3 Amendment composition and production influences mineralization in soil and total $CO_2$ emissions*). We have also changed the conclusion, to be in accordance with the discussion. The figures have been improved in term of quality and all the abbreviations have been explained in the legend.

Best regards,

Justine Barthod

[revised manuscript text omitted]